



# Metabolic alkalinity release from large port facilities (Hamburg, Germany) and impact on coastal carbon storage

Mona Norbisrath[1,2], Johannes Pätsch[1,3], Kirstin Dähnke[1], Tina Sanders[1], Gesa Schulz[1,4], Justus E. E. van Beusekom[1], Helmuth Thomas[1,2]

[1]Helmholtz-Zentrum Hereon, Institute of Carbon Cycles, Geesthacht, 21502, Germany
[2]Carl von Ossietzky University Oldenburg, Institute for Chemistry and Biology of the Marine Environment, Oldenburg, 26129, Germany
[3]University Hamburg, Institute of Oceanography, Hamburg, 20146, Germany
[4]University Hamburg, Institute of Geology, Center for Earth System Research and Sustainability (CEN), Hamburg, 20146, Germany

*Correspondence to*: Mona Norbisrath (mona.norbisrath@hereon.de)

## Abstract

Metabolic activities in estuaries, especially these of large rivers, exert profound impact on downstream coastal biogeochemistry. Here, we unravel the contribution of large industrial port facilities to these impacts and show that metabolic activity in the Hamburg port (Germany) increases total alkalinity (TA) and dissolved inorganic carbon (DIC) runoff to the North Sea. We explained this activity to be fueled by the imports of particulate inorganic and organic carbon (PIC, POC) and particulate organic nitrogen (PON) from the upstream Elbe River, resulting in maximum 90 % TA generation due to $CaCO_3$ dissolution in the entire estuary. The remaining 10 % can be attributed to a TA generation by anaerobic metabolic processes such as denitrification of remineralized PON, or other pathways. The Elbe Estuary as a whole adds approximately 15 % to the overall DIC and TA runoff. Both the magnitude and partitioning among these processes appear to be sensitive to climate and anthropogenic changes, and affects coastal $CO_2$ storage capacity.

## 1 Introduction

Human activities have increased nutrient and organic matter (OM) fluxes to the coastal ocean e.g. (Howarth et al., 1996). Estuaries play an important role in modifying these fluxes for instance by retaining part of the nitrogen fluxes through denitrification (Howarth et al., 2011;Frankignoulle et al., 1998;Seitzinger, 1988;Smith and Hollibaugh, 1993). Since estuaries have an intense biogeochemical cycling, the outer estuary typically acts as a sink of $CO_2$; the inner estuary is often a source (Borges et al., 2006;Cai and Wang, 1998;Frankignoulle et al., 1998).

Deep estuaries and semi-enclosed seas, such as the Gulf of St. Lawrence or the Baltic Sea, are mostly permanently stratified, which means that they have a strong memory effect due to ventilation time scales of the subsurface waters that are beyond the annual scale. Such waterbodies thus have a high storage capacity for carbon species such as DIC and nutrients e.g. phosphorus,





and may hold oxygen deficits (Mucci et al., 2011) leading to high N retention (De Jonge et al., 1994). Compared to deep estuaries, shallow ones like the Elbe Estuary, are usually well ventilated (Pein et al., 2021;Abril et al., 2002), have a strong benthic-pelagic coupling due to vertical exchange, and respond directly to seasonal forcing.

The Elbe Estuary is located in the northern part of Germany and is about 140 km long. It encompassing an area that begins at
the weir in Geesthacht (Elbe stream km 586), crosses the port of Hamburg (Elbe stream km 623) and discharges near Cuxhaven (Elbe stream km 727) into the North Sea, a semi-enclosed shelf sea of the eastern North Atlantic (Fig. 1). The North Sea influences the estuary with its strong tidal cycles, resulting in a semi-diurnal tidal range of 3.6 m in the port of Hamburg (Amann et al., 2012), the third largest port in Europe, and provides a continuous exchange of fresh and marine water.

Eutrophication (e.g. from agricultural fertilizers and wastewater) can cause large phytoplankton blooms both in rivers and in
the coastal zone (Hardenbicker et al., 2016). Whose decay increases oxygen consumption (Spieckermann et al., 2021;Schöl et al., 2014), and may lead to declining oxygen levels in bottom water, and possibly hypoxia in estuaries and coastal zones (Große et al., 2016;Howarth et al., 2011;Mucci et al., 2011;Thomas et al., 2009;Frankignoulle et al., 1996;Nixon, 1995). Just upstream of the port of Hamburg, dredging activities have increased the depth from around 5 m to about 20 m to guarantee accessibility of large seagoing vessels. Here, a hotspot of organic matter turnover exists (Pein et al., 2021;Sanders et al., 2018;Schöl et al.,
2014). Increasing depths favor seasonal stratification, and sedimentary structures in the port area favor the formation of oxygen-poor or even hypoxic zones (Pein et al., 2021;Kerner, 2007), which are essential for anaerobic metabolic processes that use terminal electron acceptors other than $O_2$ (e.g. $NO_3^-$, $Fe_3^+$, $Mn_4^+$, $SO_4^{2-}$) to respire organic matter. These metabolic processes release alkalinity in varying stoichiometries (Hu and Cai, 2011;Wolf-Gladrow et al., 2007;Chen and Wang, 1999;Brewer and Goldman, 1976). The resulting reduced products (e.g. $N_2$, $H_2S$, $Fe_2^+$) can be transported back into the water
column. If such reduced products, with the exception of $N_2$, can be reoxidized in the water column, any increase in alkalinity from the generation will be consumed. Reduced products can escape reoxidation via permanent burial (e.g. $FeS_2$ (pyrite)), or via escape to the atmosphere (e.g. $N_2$). In addition to anaerobic processes, dissolution of calcium carbonate ($CaCO_3$) also generates alkalinity in a TA:DIC ratio of 2:1, which can be reversed by precipitation (Chen and Wang, 1999). For the purpose of this paper, we consider $CaCO_3$ dissolution as a metabolic process favored by lower pH, e.g. bacterial degradation of organic
matter, leading to generation of alkalinity. Earlier studies (Francescangeli et al., 2021;Kempe, 1982) confirm the hydro-chemical conditions favoring $CaCO_3$ dissolution in the lower Elbe Estuary.

We shed light on the biogeochemical cycling of TA, DIC and in particular nitrate and employ an approach that combines observational and modeling techniques (Schwichtenberg et al., 2020). We want to answer the questions of how much metabolic alkalinity is released from the Elbe Estuary into downstream coastal waters of the North Sea, and how their $CO_2$ uptake changes
under altered metabolic alkalinity inputs as a consequence of climate and anthropogenic changes.



## 2 Methods

### 2.1 Study site and observed parameters

#### 2.1.1 Study site

This study based on samples that we collected during a cruise (LP20190603) on RV *Ludwig Prandtl* on two consecutive days in June 2019 during ebb tide. We sampled from the German Bight around the island of Scharhörn upstream to Oortkaten situated in the riverine part of the estuary (Fig. 1). We took surface water (1.2 m depth) samples every 20 minutes out of the flow-through FerryBox system (Petersen et al., 2011) that concurrently provides the required physical parameters such as salinity, temperature and oxygen. Water depth in the Elbe Estuary was around 15-20 m but sharply decreased to 5 m upstream

Elbe stream km 620. For mass balance calculations, we separated the Elbe Estuary in 6 boxes, indicated by the lines in Fig. 1, of which box 1 (upstream the Hamburg port) and box 6 (North Sea) act as boundary conditions.

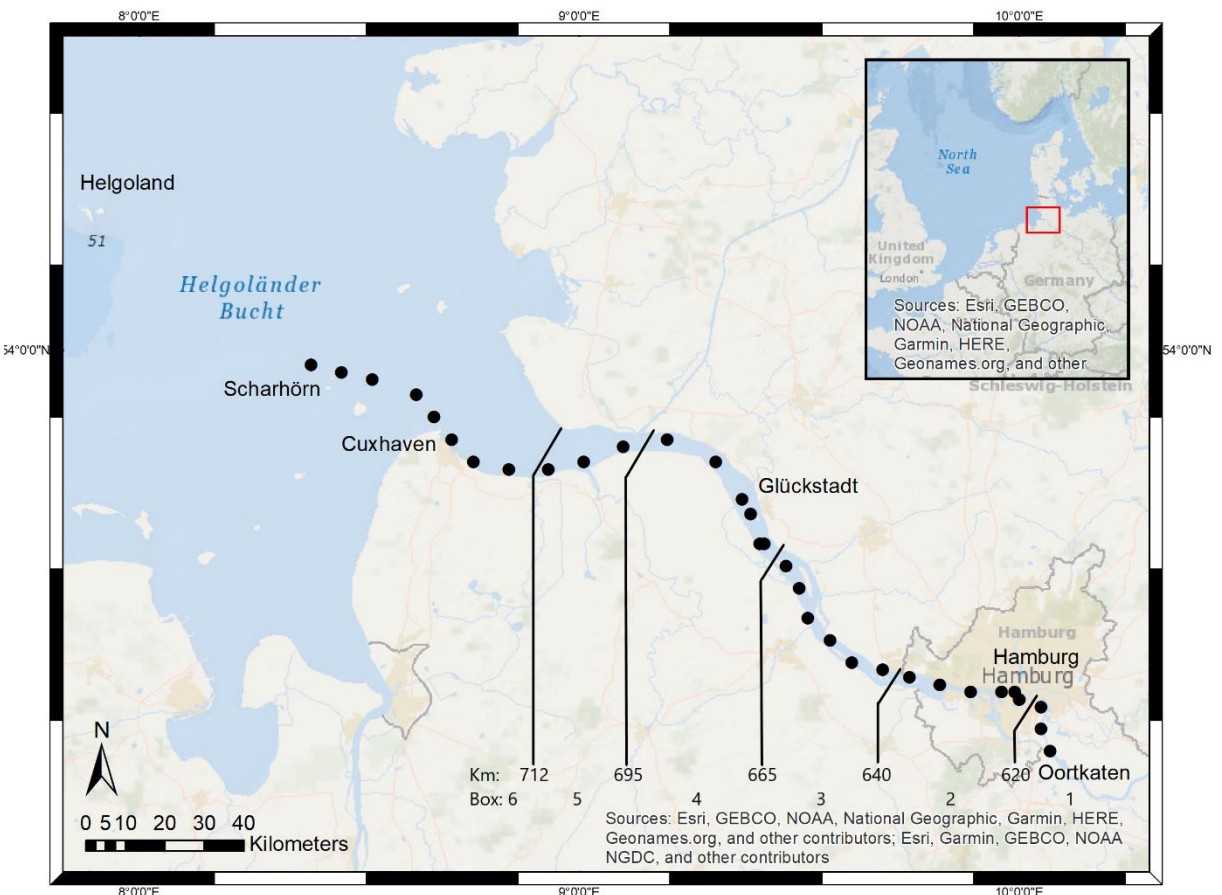



**Figure 1. Study site.** The Elbe Estuary with sampling stations in June 2019 (dots) and the spatial separation of the estuary for the mass balance calculation (lines), whereby box 2 indicates the port of Hamburg in the upper estuary, box 3 and box 4 in the middle estuary, and
box 5 in the lower estuary.

### 2.1.2 TA and DIC

First, water samples for total alkalinity (TA) and dissolved inorganic carbon (DIC) measurements were filled in 300 mL BOD (biological oxygen demand) bottles and preserved with 300 µL saturated mercuric chloride ($HgCl_2$) to stop biological activity. We sealed the bottles with ground-glass stoppers, Apiezon® type M grease, free of air bubbles and protected them against
opening with plastic caps. Samples were stored in a cool place in the dark until measured.

We performed the measurements of TA and DIC at Helmholtz-Zentrum Hereon using a VINDTA 3C (Versatile INstrument for the Determination of Total dissolved inorganic carbon and Alkalinity, marianda - marine analytics and data). VINDTA 3C measures TA by potentiometric and DIC by coulometric titration, respectively (Shadwick et al., 2011). We used certified reference material (CRM batch # 187) provided by A. G. Dickson (Scripps Institution of Oceanography) to calibrate TA and
DIC measurements and ensured a precision of $\pm$ 2 µmol kg$^{-1}$.

### 2.1.3 Nutrients and stable nitrate isotopes

Samples for nutrients and stable isotopes of nitrate ($NO_3^-$) were collected concurrently with TA and DIC samples, to measure the concentration and the isotopic composition of nitrate respectively. We collected three 15 mL Falcon tubes for triplicate nutrient measurements, and one 100 mL PE bottle (acid-washed overnight with 10 % HCl) for stable isotope analysis. Samples
were filtered through pre-combusted GF/F filters, and then frozen for onshore analyses.

We used a continuous flow automated nutrient analyzer (AA3, SEAL Analytical) and a standard colorimetric technique (Hansen and Koroleff, 2007) to measure concentrations of dissolved nitrate ($NO_3^-$), nitrite ($NO_2^-$) and phosphate ($PO_4^{3-}$), and ammonium ($NH_4^+$) with a fluorometric method (Kérouel and Aminot, 1997), in triplicate.

We applied the denitrifier method (Casciotti et al., 2002; Sigman et al., 2001) to determine the stable isotope ratios of $\delta^{15}N$ and
$\delta^{18}O$ of nitrate. We used denitrifying *Pseudomonas aureofaciens* (ATCC#13985), which lacks nitrous oxide reductase activity. The bacteria reduced nitrate and nitrite in the filtered water sample to nitrous oxide ($N_2O$). The $N_2O$ produced was measured using a GasBench II coupled to an isotope mass spectrometer (Delta Plus XP, Fisher Scientific). Concurrently, we used two international standards (USGS34, $\delta^{15}N\text{-}NO_3^- = -1.8$ ‰, $\delta^{18}O\text{-}NO_3^- = -27.9$ ‰; IAEA, $\delta^{15}N\text{-}NO_3^- = +4.7$ ‰, $\delta^{18}O\text{-}NO_3^- = +25.6$ ‰) and one internal standard ($\delta^{15}N\text{-}NO_3^- = +7.6$ ‰, $\delta^{18}O\text{-}NO_3^- = +24.4$ ‰) for data correction in each run. The standard
deviation for standards and samples was <0.2 ‰ (n = 4) for $\delta^{15}N\text{-}NO_3^-$ and <0.5 ‰ (n = 4) $\delta^{18}O\text{-}NO_3^-$. The nitrite concentration of the samples was usually less than 5 %. When it exceeded this threshold, nitrite in the samples was removed with Sulfamic Acid (4 % Sulfamic Acid in 10 % HCl) prior to analysis (Granger and Sigman, 2009).

Variations in the natural abundance of stable isotopes are represented as relative differences in isotope ratios. The isotope ratio R is the ratio of heavy to light isotopes. Since isotope differences are very small, the delta-$\delta$-notation is used to describe the



isotopic composition of samples (Eq. 1). Therefore, the isotopic ratio of a sample ($R_{\text{sample}}$) is given relative to the ratio of an internationally accepted reference material ($R_{\text{reference}}$). Atmospheric $N_2$ and Vienna Standard Mean Ocean Water (VSMOW) are the reference materials for nitrogen and oxygen, respectively. The delta-δ-notation is calculated as follows:

$$\delta\ (\text{‰}) = \left( \frac{R_{\text{sample}} - R_{\text{reference}}}{R_{\text{reference}}} \right) \times 1000 \tag{1}$$

We identified the nitrate sources river upstream from the minimum isotope values at Elbe stream km 705 using Eq. 2. We

focused on the upper fresh water part of the estuary to identify the isotope values and the source of the additional nitrate ($\delta_{\text{ad}}$) added to the estuary between Elbe stream km 609 and 705. We used a simple mixing model (Sanders et al., 2018) with the $\delta^{15}N$ and $\delta^{18}O$ values and associated nitrate concentrations ($C$):

$$\delta_{\text{ad}} = \frac{((\delta_{705}\ x\ C_{705}) - (\delta_{609}\ x\ C_{609}))}{(C_{705} - C_{609})} \tag{2}$$

### 2.1.4 Ancillary parameters

We used pre-combusted (4 h, 450 °C) GF/F filters to sample for particulate organic carbon (POC) and particulate organic nitrogen (PON) concentrations. First, the filters were measured for total carbon (TC) as a bulk sample. Then, the filters were acidified 3 times with 1 N HCl and measured again for POC. The particulate inorganic carbon (PIC) was derived from TC-POC. The filters were measured with a CHN-elemental analyzer (Eurovector EA 3000, HEKAtech GmbH) in the Institute of Geology, University Hamburg. The measurements were calibrated against the standards Sulfanilamide (C = 41.84 %, N =

16.27 %) und Acetanilide (C = 71.09 %, N = 10.36 %).

The oxygen data were directly measured with the FerryBox, and calibrated with reference to parallel samples analyzed using the Winkler method.

The water depths that we used for the mass balances (see below) were measured during the cruise with the on board Acoustic Doppler Current Profiler (WorkHorse Broadband ADCP 1200kHz, Firmware Version 51.40.) (Cysewski et al., 2018).

### 2.2 Gas flux calculation

As an input function for the mass balances (see below), we estimated $CO_2$ and $O_2$ gas exchange ($F$) between the atmosphere and water. We used the following equation:

$$F_{\text{gas}} = k_{\text{gas}}\ x\ (d_{\text{gas}}) \tag{3}$$

where $k_{\text{gas}}$ is the transfer velocity of the gases ($O_2$ and $CO_2$), and $d_{\text{gas}}$ is the difference between the atmospheric and aquatic

concentration, calculated as follows:

$$d_{\text{gas}} = [\text{gas}]_{\text{at 100\% saturation}} - [\text{gas}]_{\text{observed}} \tag{4}$$

The $CO_2$ concentration of the sample (observed) was computed from DIC, TA, salinity and temperature using the $CO_2SYS$ program (Lewis and Wallace, 1998) and the dissociation constants for fresh water (Millero, 1979). The $CO_2$ concentration of the sample corresponds the mentioned $pCO_2$. The $CO_2$ concentration (at 100 % saturation) was obtained from TA and the



global atmospheric $CO_2$ saturation of $416 \pm 0.13$ µatm (Dlugokencky and Tans, 2021) by using the same previously mentioned program and constants.

The transfer velocity was calculated after Wanninkhof (2014) as follows:

$$k_{\text{gas}} = \frac{0.251\, x\, (U_{10})^2}{(\frac{Sc}{Sc_{\text{ref}}})^{-0.5}} \tag{5}$$

where $0.251$ is the coefficient of gas transfer, $U_{10}$ is the wind speed in m sec $^{-1}$ measured at 10 m height (provided by DWD (2020), $Sc$ is the Schmidt number, the kinematic viscosity of water divided by the diffusion coefficient of the gas (Wanninkhof, 2014) and $Sc_{\text{ref}}$ is the $Sc$ value relative to the reference conditions of the gas at 20 °C in fresh water, which is 510 for $O_2$ and 600 for $CO_2$. The uncertainty of the flux calculation has been estimated to 20 % (Wanninkhof, 2014;Watson et al., 2009).

**2.3 Mass balances**

**2.3.1 Box model approach**

To balance the net dissolved budgets of TA, DIC, $NO_3^-$ and $O_2$ in the Elbe Estuary, we used a box model approach (Fig. 2) based on our observations. The box model approach allowed us to distinguish conservative contribution from marine and fresh water end-members based on salinity (i. e. baseline), the input and output values as advected transport, and the atmospheric gas exchange ($O_2$ and $CO_2$), of the observed parameters TA, DIC, $NO_3^-$ and $O_2$ between and within boxes. The resulting difference is the closing term given as net dissolved budgets, and referred to here as metabolic gains. To attribute the potential processes affecting metabolic generation, we distinguished between pelagic generation associated with aerobic conditions, i.e., nitrification, which consumes TA, and benthic generation corresponding to anaerobic conditions, where anaerobic metabolic processes generate TA. We defined 6 boxes based on spatial and salinity considerations, with box 1 representing the fresh water (upstream the Hamburg port) and box 6 the coastal ocean (North Sea). Both boxes, 1 and 6 determined the start and end, and were not balanced while acting as boundary conditions. Calculation and end-member properties are shown in Table 1.

Discharge per box was calculated using the observed depths, in combination with an average river discharge value (provided by FGG (2021)) measured from the last tide-free, long term monitoring station Neu Darchau (Elbe stream km 536). Accordingly, we computed the fill time, or in other words the fresh water flushing time of each box, as box volume divided by discharge (Table 1).

We used the following equation to calculate the metabolic gains by assuming a steady state:

$$\frac{\delta C}{\delta t} = 0 = F_{\text{Input}} + F_{\text{Output}} + F_{\text{Baseline}} + F_{\text{ASF (O2,CO2)}} + F_{\text{Metabolic}} \tag{6}$$

where $F_{\text{Input}}$ is the input value, $F_{\text{Output}}$ the output value, $F_{\text{Baseline}}$ the salinity corrected baseline value, $F_{\text{ASF (O2, CO2)}}$ the air-sea flux of $O_2$ and $CO_2$, respectively, and $F_{\text{Metabolic}}$ the metabolic gain as the closing term.



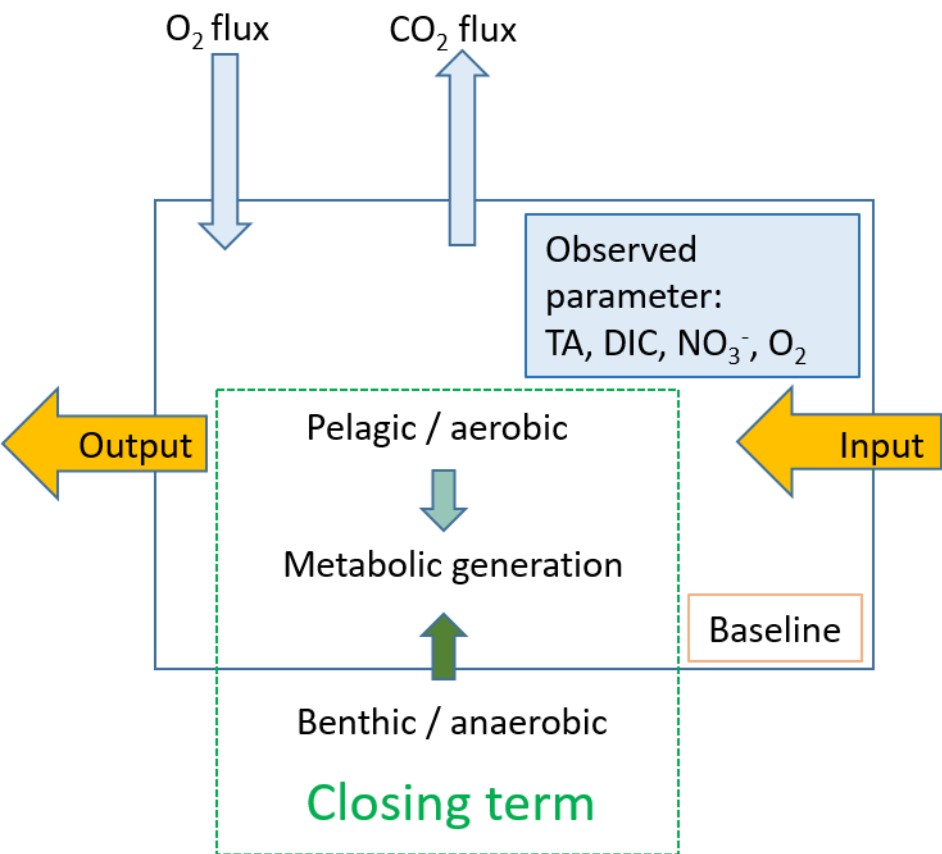

**Figure 2. Schematic mass balance approach.** We used a box model to balance the net dissolved budgets along the Elbe Estuary. The
observed parameters are shown in light blue, the closing term is set as metabolic generation (green), which is fueled by pelagic, i.e. aerobic,
and benthic, i.e. anaerobic processes, but without allocating how much of what is generated where. The input and output is shown in yellow,
and the latter acts as the input for the downstream box. To complete DIC and $O_2$, we included the atmospheric gas exchange. The baseline
is calculated assuming conservative mixing of fresh and marine end-members.

**Table 1. Elbe Estuary data for the mass balance calculation.** The water depths are measured data of the Acoustic Doppler Current Profiler
(ADCP) on board of RV *Ludwig Prandtl*, and the river length is based on the Elbe stream km. The river width is based on measurements
with Google Maps®, and we used a standardized river width of 2 km for box 3, 4 & 5. The average river discharge on our sampling days
was 423 $m^3$ $sec^{-1}$ (provided by FGG (2021)) measured at the last tide-free long term sampling station in Neu Darchau (Elbe stream km 536).
Average box values are given for total alkalinity (TA), dissolved inorganic carbon (DIC), nitrate ($NO_3^-$) and salinity. For the outer boundary
conditions (box 6), we used a maximum salinity of 33.26, and respective TA (2449 µmol $kg^{-1}$), DIC (2132 µmol $kg^{-1}$), and $NO_3^-$ (0.1 µmol
$L^{-1}$) values taken on cruise HE541 (54.062456° N, 8.015919° E), which took place two months later in September 2019 and represents
average summer time values for the German Bight. Standard deviation ± sd as spatial variability is given when possible.



| Species | Unit | Box 1 | Box 2 | Box 3 | Box 4 | Box 5 | Box 6 |
|---|---|---|---|---|---|---|---|
| Elbe stream km | km | 607 - 619 | 619 - 639 | 639 - 666 | 666 - 693 | 693 - 710 | - |
| Stream with | km | 0.3 | 0.5 | 2.0 | 2.0 | 2.0 | - |
| Stream length | km | 12 | 20 | 27 | 27 | 17 | - |
| Depth | km | 0.00493 | 0.01392 | 0.016056 | 0.011999 | 0.016926 | - |
| Water volume | km$^3$ | 0.0177 | 0.1392 | 0.8670 | 0.6479 | 0.5755 | - |
| Fill time | d | 0.4865 | 3.8094 | 23.7241 | 17.7304 | 15.7468 | - |
| av. TA ± sd | µmol kg$^{-1}$ | 1289 ± 8 | 1512 ± 40 | 1630 ± 42 | 1733 ± 35 | 1879 ± 21 | 2449 |
| av. DIC ± sd | µmol kg$^{-1}$ | 1288 ± 30 | 1589 ± 58 | 1701 ± 39 | 1793 ± 29 | 1896 ± 4 | 2132 |
| av. NO$_3^-$ ± sd | µmol L$^{-1}$ | 74 ± 0 | 91 ± 6 | 114 ± 7 | 139 ± 9 | 161 ± 6 | 0.1 |
| av. Salinity ± sd | - | 0.41 ± 0.0 | 0.45 ± 0.0 | 0.61 ± 0.1 | 1.07 ± 0.4 | 5.17 ± 1.4 | 33.26 |

The uncertainty of the closing term has been estimated by using the analytical measurement precision of 2 µmol kg$^{-1}$ for DIC and TA, 0.5 µmol L$^{-1}$ for NO$_3^-$, and 5 % for O$_2$ (Petersen et al., 2011). The gas exchange was considered with an uncertainty of 20 % (Wanninkhof, 2014;Watson et al., 2009). The river discharge uncertainty was 5 % (Léonard et al., 2000) and was added to the error in the end. The analytical measurement precision of POC and PON was 0.05 % and 0.005 %, respectively (Gaye et al., 2022), calculated from the average box 1 values. The analytical errors were calculated with the following equation per parameter and box:

$$X = (\sum_i x_i^2)^{0.5} \tag{7}$$

where $X$ is the combined error and $x_i$ are the errors of the individual observations.

### 2.3.2 TA source attribution

In order to assess the total alkalinity gains in the entire Elbe Estuary, we compared the mass balance gains with measured riverine POM properties.

By using the imported riverine PIC and POC, we derived the maximum amount TA fueled by CaCO$_3$ dissolution as source, and estimated the remaining amount of PIC transported downstream.

In order to estimate the TA gain that can be fueled by CaCO$_3$ dissolution, we considered the observed metabolic DIC, the average imported PIC:TC (particulate inorganic carbon:total carbon) ratio (0.29 ± 0.05), and the TA:DIC ratio for CaCO$_3$ dissolution (2) (Chen and Wang, 1999). To get the remaining DIC generated by e.g. organic matter respiration (POC), we multiplied the metabolic DIC gain by 0.71 (1 – 0.29 = 0.71). To arrive at the corresponding TA that was not fueled by CaCO$_3$ dissolution, we subtracted the TA fueled by CaCO$_3$ dissolution from the entire metabolic TA gain. We performed all calculations per box.

In order to estimate the amount of PIC that is transported through the estuary and not used for TA generation by CaCO$_3$ dissolution, we used the imported PIC and the previously calculated corresponding TA. The following boxes then do not refer





to the imported PIC, but to the product of the previous box that correspond to the unused transported PIC. In case of negative values such as in box 4, we set the transported PIC in the following calculation to 0.

For the coupling of carbon and nitrogen, we used the imported PON to estimate the amount that is transported unused through the Elbe Estuary, and to estimate whether or not riverine PON income is sufficient to generate $NO_3^-$ and TA.

For estimating the TA generation fueled by denitrification that we attribute here to imported riverine PON, we calculated the
205 amount of PON that is available for denitrification by subtracting the nitrified PON (i.e. entire estuarine metabolic $NO_3^-$ gain) of the imported PON (box 1). With the latter calculated PON available for denitrification, the ratio of TA:DIC resulting from denitrification (0.9) (Chen and Wang, 1999), and the entire TA gain that is not fueled by PIC, we calculated the possible TA gain of PON that could be generated in the estuary without the occurrence of nitrification.

To estimate the amount of PON that can be transported downstream, we used the imported PON, the sum of the $NO_3^-$ gain of
210 the mass balances, the previously calculated TA gain that was not fueled by PIC, and the TA:DIC ratio (0.9) for denitrification. The following boxes then do not refer to the imported PON, but to the product of the previous box. In case of negative values such as in box 4, we set the transported PON in the following calculation to 0.

## 2.4 Biogeochemical simulations

For estimating the effects of metabolic produced alkalinity in the estuary on the North Sea and the continental shelf, we applied
the 3D-ECOHAM model (Schwichtenberg et al., 2020). Pätsch and Kühn (2008) first described the ECOHAM model domain for this study (see their Fig. 1). Meterological forcing for both models were derived from the ERA5 reanalysis dataset (Hersbach et al., 2020).

The physical parameters temperature, salinity, horizontal and vertical advection as well as turbulent mixing were calculated using the hydrodynamic model HAMSOM (Backhaus, 1985). This is a baroclinic primitive equation model using the
220 hydrostatic and Boussinesq approximations. Details are described by Backhaus and Hainbucher (1987) and Pohlmann (1996). We run the hydrodynamic model prior to the biogeochemical part. The daily result fields were saved to run the biogeochemical model in offline mode.

The relevant biogeochemical processes and their parameterizations have been detailed in Lorkowski et al. (2012). TA and DIC are computed prognostically. The pelagic biogeochemical part is driven by planktonic production and respiration, calcite
formation and dissolution, pelagic and benthic degradation and remineralization, and also by atmospheric deposition of reactive nitrogen. All of these processes affect TA. The air-sea flux of $CO_2$ was calculated for the North Sea region between 51° N and 59.5° N according to (Wanninkhof, 2014).

We extracted the year 2001 from the simulation run 1979-2014 for the analysis in this paper. Four different scenarios (50, 86, 100, and 150 % TA load) were run, with the 100 % scenario being the reference scenario with full riverine TA and DIC input.
In comparison, the 86 % scenario reflects river input without the metabolic alkalinity generation corresponding to our calculated metabolic TA generation of 14 % throughout the Elbe Estuary. We run two other scenarios, one with a reduced TA load (50 %) and one with an increased TA load (150 %) for a broader comparison. Daily data of fresh water fluxes from 254





rivers discharging into the North Sea were used (Große et al., 2017). Corresponding river load data of nutrients, organic matter, DIC, and TA were applied.

## 3 Results and Discussion

### 3.1 Salinity, TA, DIC, and nutrient distribution along the estuary

Along the transect through the Elbe Estuary, the salinity varied from 0.40 in the fresh water part (Elbe stream km 609-680), to 27.84 in marine waters in the German Bight. The strongest salinity gradient was observed between Elbe stream km 683 and 715 (box 5, tidal front) with an increase of 10 (Fig. 3a). The limited intrusion in salinity in the fresh water part visible in Fig. 3a restricted the clear subdivision of the boxes 1 to 3 in Fig. 3b.

TA and DIC increased from the upper estuary (Elbe stream km 609) to the mouth of the estuary (Fig. 3a). The lowest TA and DIC concentrations were 1281 µmol TA kg$^{-1}$ and 1256 µmol DIC kg$^{-1}$ at 0.41 salinity in Oortkaten, the highest were 2272 µmol TA kg$^{-1}$ and 2016 µmol DIC kg$^{-1}$ at 27.84 salinity around Scharhörn. TA was higher than DIC in marine and brackish water, but in the fresh water part of the estuary, DIC was higher than TA. At a salinity of 6.96, TA and DIC were equal with 1922 µmol kg$^{-1}$. The strongest increase in TA is observed in the fresh water part close to 0.40 salinity, between Elbe stream km 609 and extending to Elbe stream km 670 (Fig. 3a). In the outer estuary, TA and DIC increased almost proportional to the salinity gradient without obvious biological impact (Fig. 3a,b).

Nitrate ($NO_3^-$) concentrations increased from 73 µmol $NO_3^-$ L$^{-1}$ near Oortkaten to a maximum of 165 µmol $NO_3^-$ L$^{-1}$ in the lower estuary at a salinity of 5.08 (Elbe stream km 705, box 5) (Fig. 3a), which is downstream of the maximum turbidity zone (MTZ is located between Elbe stream km 664-670, not shown). Further, nitrate mixed conservatively with low $NO_3^-$ seawater to 15 µmol $NO_3^-$ L$^{-1}$ at km 750.

Ammonium ($NH_4^+$) reached a pronounced maximum of 15 µmol $NH_4^+$ L$^{-1}$ in the Hamburg port area, upstream of the $NO_3^-$ maximum, while varying in a range around 1 µmol $NH_4^+$ L$^{-1}$ in the remainder of the estuary (Fig. 3a). This maximum coincides with the strongest gradients in DIC and TA and can be attributed to high remineralization rates in the port area, as is also evidenced by high apparent oxygen utilization (AOU) (Fig. 3a).

Phosphate ($PO_4^{3-}$) trends are comparable to nitrate, with a sharp increase in the Hamburg port area and a maximum concentration of 2 µmol $PO_4^{3-}$ L$^{-1}$ in the lower estuary (box 5). Similar to TA and DIC, $PO_4^{3-}$ increased where the Elbe River enters the port area.

The $\delta^{15}N$-$NO_3^-$ and $\delta^{18}O$-$NO_3^-$ isotopic values were higher in the upper estuary and decreased downstream along the estuary to their respective minima at km 705, coinciding with the $NO_3^-$ maximum. The $\delta^{15}N$ values ranged from 12.4 ‰ to 17.1 ‰, the $\delta^{18}O$ values from 5.1 ‰ to 9.1 ‰ in the fresh water part.

The $\delta^{15}N$ values of suspended particulate matter (SPM) ($\delta^{15}N_p$) showed a pronounced increase from 6 to 10 ‰ in box 2, indicating an enrichment of heavy isotopes in the particulate matter due to the degradation of particulate organic matter (POM). This strong increase coincides with the ammonium peak and the declining $\delta^{15}N$-$NO_3^-$.

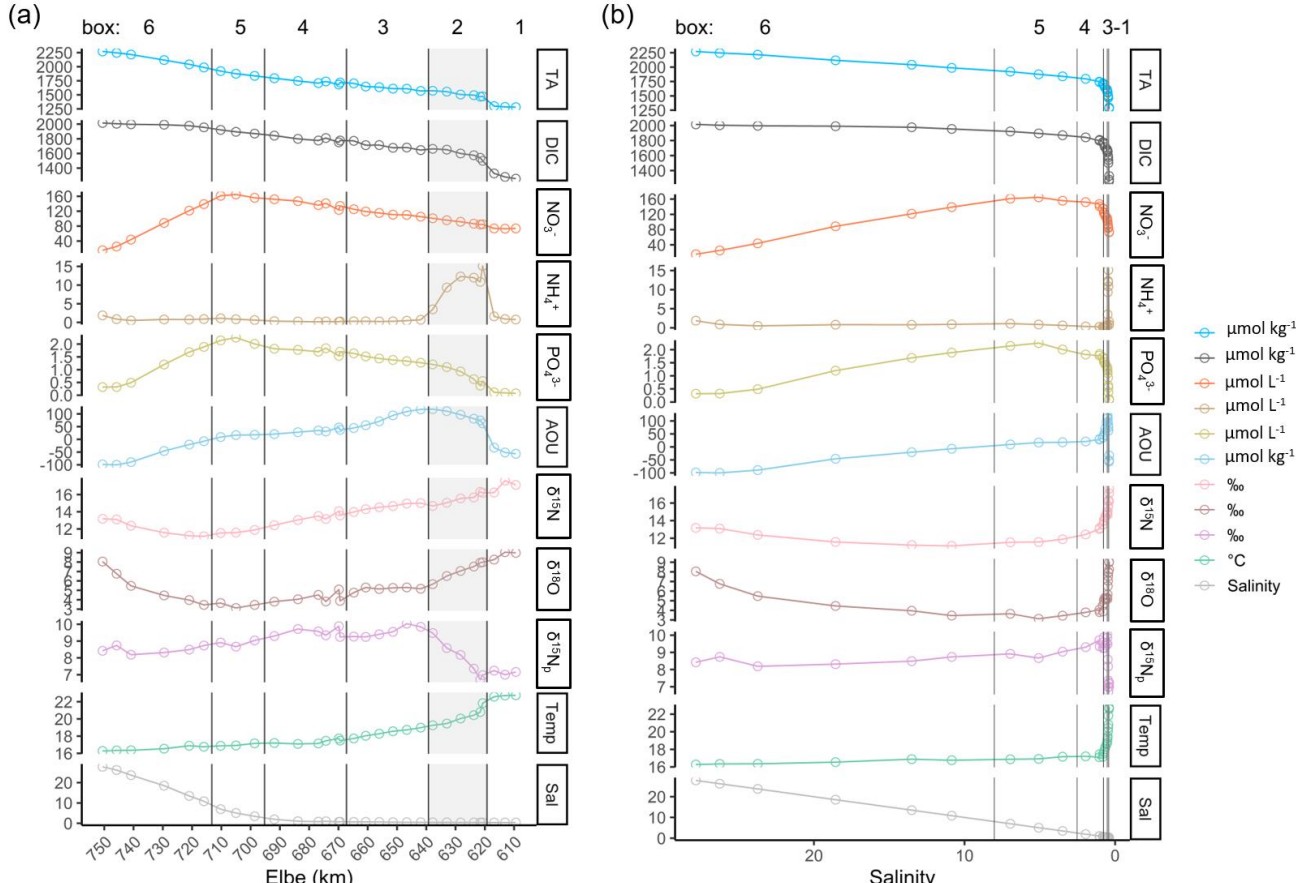

**Figure 3. Spatial and salinity dependent observed parameter distribution along the Elbe Estuary.** The data are from two days in June 2019. (a) Spatial observed parameter distribution from the inner estuary (Elbe stream km 609, right side) to the German Bight around the island of Scharhörn (computed Elbe stream km 750, left side). Total alkalinity (TA), dissolved inorganic carbon (DIC), nitrate ($NO_3^-$), ammonium ($NH_4^+$), phosphate ($PO_4^{3-}$), apparent oxygen utilization (AOU), delta values of nitrate isotopes ($\delta^{15}N$) and ($\delta^{18}O$) of filtered water, and suspended particulate matter ($\delta^{15}N_p$), temperature (Temp) in $^\circ$ C, and salinity (Sal) are shown. The vertical lines indicate the designated boxes for the mass balance calculation, in which the inner boundary (box 1) is on the right side of the plot, followed by the shaded box 2 with the Hamburg port area (upper estuary), boxes 3 and 4 (middle estuary), 5 (lower estuary), and the outer, marine boundary (box 6) on the left side of the plot. (b) Salinity dependent observed parameter distribution. Note the different y-axis scales within the different panels of both plots.

## 3.2 Metabolic activity in the port

The North Sea's carbonate system is affected by coastal areas (Schwichtenberg et al., 2020), necessitating a specific investigation of estuarine based TA generation and an estimation of its impact on the oceanic $CO_2$ uptake capacity. In order to determine the TA, DIC, and $O_2$ related metabolic activity in the Elbe Estuary, we estimated the air-sea gas exchange of $CO_2$



and $O_2$ as these are non-advective processes too (Fig. 4a, Table 2) along the Elbe Estuary. Upstream of the port of Hamburg,
the $pCO_2$ was slightly supersaturated (465 µatm $CO_2$) relative to the atmosphere (416 µatm $CO_2$). Highest values of 2074 µatm
$CO_2$ in Hamburg port water indicate strong $CO_2$ degassing. Further downstream, the $pCO_2$ decreased again (Table 2). In other
European estuaries a similar $pCO_2$ pattern was recorded (Frankignoulle et al., 1998)[4]. $pO_2$ mirrored the $pCO_2$ evolution along
the estuary and both together allowed us to estimate respective sea to air (in case of $CO_2$) and air to sea (in case of $O_2$) fluxes
(Fig. 4a).

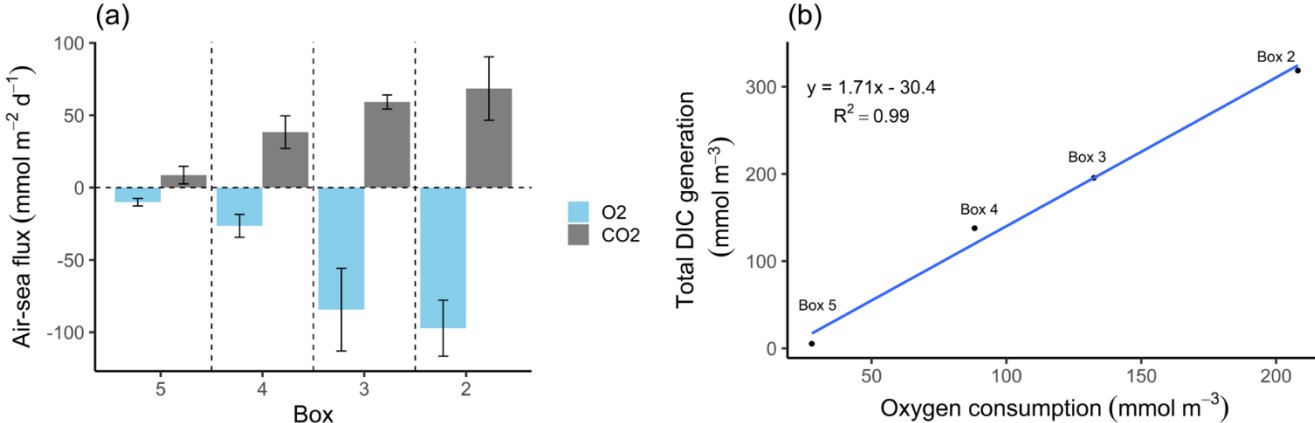

**Figure 4. $O_2$ and $CO_2$ characteristics.** (a) Air-sea fluxes of oxygen ($O_2$) and carbon dioxide ($CO_2$) per box, calculated from $pO_2$ and
observed oxygen concentrations, and from $pCO_2$ values and observed TA, respectively (Table 2). Values are shown with standard deviation
represented as spatial variability (error bars). Positive values indicate outgassing, negative values a gas uptake into the water. The Hamburg
port area is located in box 2, the lower estuary in box 5. (b) Relationship between metabolic DIC generation and $O_2$ consumption along the
estuary in concentration (mmol m$^{-3}$). Please note the different parameters in the panels.

The highest metabolic DIC generation and $O_2$ consumption (Fig. 4a,b, Table 2) occurred in the port of Hamburg, where most
of the $CO_2$ produced remained dissolved, with only a moderate amount being emitted to the atmosphere. The strong metabolic
activity is also reflected by the high oxygen influx, driven by low oxygen conditions in the port of Hamburg (Table 2, see for
example Amann et al. (2012) & Schöl et al. (2014)). The high slope of 1.71 (Fig. 4b) of DIC and $O_2$ exceeds the ratio of 0.5
to 1 (e.g.Thomas (2002)), indicating anaerobic or inorganic DIC sources. The remnants of the high metabolic activity in box
2 are transported further downstream to eventually approach equilibrium in the lower estuary (box 5, Fig. 4a,b). Accordingly,
here the observed metabolic values obtained for DIC (Fig. 5) and $O_2$ (AOU, Fig. 3) constitute to a large degree advected
signals.


**Table 2. Average calculated gas values per box.** $O_2$ concentration is the observed oxygen. $O_2$ consumption is the amount of oxygen
consumed (i.e. AOU + oxygen uptake). Metabolic DIC gen. is the generated DIC. $O_2$ ASF is the oxygen, and $CO_2$ ASF is the carbon dioxide
air-sea flux along the surface. Positive flux values indicate outgassing, and negative values an absorption into the water. $pCO_2$ is the partial
pressure of carbon dioxide and corresponds to the $CO_2$ concentration in the water. The standard deviation ± sd as spatial variability is given





when possible. An uncertainty estimation including the uncertainties of the analytical measurements, the air-sea flux estimation, and the river discharge is given as (± absolute errors) by an error propagation (Methods).

| Species | Unit | Box 1 | Box 2 | Box 3 | Box 4 | Box 5 |
|---|---|---|---|---|---|---|
| Elbe stream km | km | 607 - 619 | 619 - 639 | 639 - 666 | 666 - 693 | 693 - 710 |
| $O_2$ concentration ± sd | mmol m$^{-3}$ | 316 ± 10 | 193 ± 16 | 210 ± 30 | 264 ± 9 | 278 ± 2 |
| $O_2$ consumption | mmol m$^{-3}$ | - | 208 (± 15.7) | 133 (± 30.3) | 88 (± 21.3) | 28 (± 20.7) |
| Metabolic DIC gen. | mmol m$^{-3}$ | - | 319 (± 5.4) | 196 (± 18.8) | 138 (± 12.5) | 6 (± 4.0) |
| $O_2$ ASF ± sd | mmol m$^{-2}$ d$^{-1}$ | 53 ± 12 | -97 ± 19 | -84 ± 28 | -26 ± 8 | -10 ± 3 |
| $CO_2$ ASF ± sd | mmol m$^{-2}$ d$^{-1}$ | 2 ± 14 | 68 ± 22 | 59 ± 5 | 38 ± 11 | 9 ± 6 |
| $pCO_2$ ± sd | µatm | 465 ± 341 | 2074 ± 488 | 1833 ± 120 | 1554 ± 210 | 713 ± 208 |

Using the mass balances, we calculated the net dissolved budgets of TA, DIC, and $NO_3^-$, showing only the net gain but no indication of the processes responsible for it. We calculated the highest metabolic gains of both TA and DIC in the Hamburg

port area (box 2), identifying this area as the one with the highest metabolic activity (Fig. 5), almost an order of magnitude larger than in the downstream boxes. Accordingly, very high metabolic fluxes, i.e. area and time normalized values, of TA (808 mmol m$^{-2}$ d$^{-1}$) and DIC (1164 mmol m$^{-2}$ d$^{-1}$) were obtained for the Hamburg port area (box 2) (Fig. 5, Table 3), with the downstream values below 150 mmol m$^{-2}$ d$^{-1}$, respectively, further diminishing toward the North Sea with negligible values in box 5. Similarly, the contribution of metabolically generated DIC is higher in the port area, than river downstream where $CO_2$

equilibration with the atmosphere gains (relative) importance. Compared to the metabolic gains, the air-sea flux (ASF) constitutes only a minor fraction of the metabolic fluxes (Fig. 5). The high release of DIC relative to TA (Fig. 5) are in line with our above result of high $pCO_2$ in box 2 (Fig. 4a).

In contrast to TA and DIC, the metabolic $NO_3^-$ fluxes have a similar magnitude along the estuary, with highest fluxes (63 mmol m$^{-2}$ d$^{-1}$) in the port of Hamburg and lower fluxes (between 16 and 21 mmol m$^{-2}$ d$^{-1}$) in the downstream parts of the estuary.

Although the relative difference between box 2 and the downstream boxes is much smaller than for TA and DIC (Fig. 5).





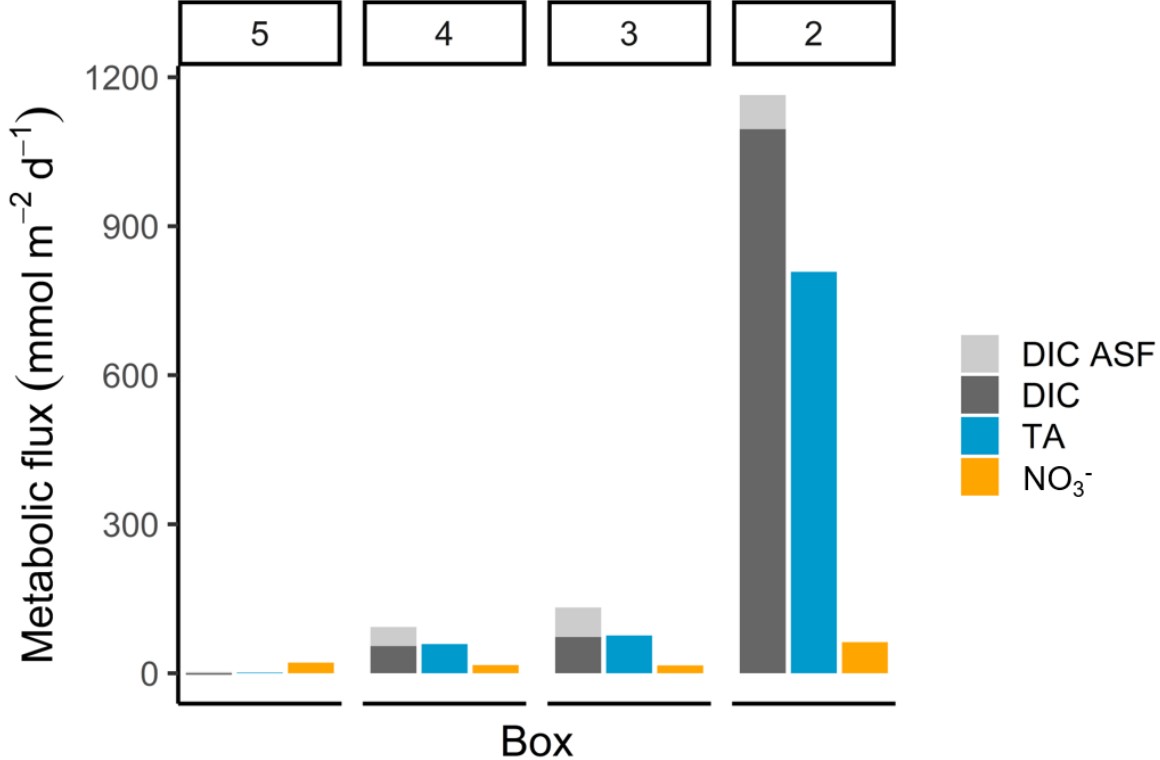

**Figure 5. Metabolic fluxes of total alkalinity, dissolved inorganic carbon, and nitrate.** The metabolic fluxes of TA (blue), DIC (dark grey) and $NO_3^-$ (orange) in mmol m$^{-2}$ d$^{-1}$ are shown, with additionally visualizing the air-sea flux (ASF) contribution to DIC (light grey) to demonstrate the entire generation and their relative magnitude.


We estimated the total dissolved metabolic DIC and TA loads from all boxes released into the Elbe Estuary in the North Sea (excluding the air-sea fluxes). Our calculations yield annual metabolic loads of 5.6 Gmol yr$^{-1}$ for TA and 6.5 Gmol yr$^{-1}$ for DIC. In relation to the most recent data from 2017 (Pätsch and Lenhart, 2019), these metabolic loads represent 16 % of the overall DIC (39.6 Gmol DIC yr$^{-1}$) and 14 % of the overall TA (40.3 Gmol TA yr$^{-1}$) Elbe loads.

**3.3 Carbon based constraints on TA generation**

In order to explain the net total alkalinity gains generated in the Elbe Estuary that were identified with the mass balances, we compared them to imported riverine particulate organic matter (POM) properties from samples taken simultaneously to our Elbe transect (Table 3). We used imported riverine particulate inorganic and organic carbon (PIC, POC) properties to attribute the TA gain fueled by calcium carbonate ($CaCO_3$) dissolution and to estimate the remaining PIC amount that is transported 335 downstream to identify a possible $CaCO_3$ deficit along the estuary.



We obtained TA generation from $CaCO_3$ dissolution referring to the metabolic, i.e. POM driven DIC generation multiplied by the observed PIC:TC ratio of $0.29 \pm 0.05$ (average of entire estuary) of the imported POM. We assumed spontaneous, i.e. maximum dissolution of $CaCO_3$, which integrated over the estuary, can contribute up to 90 % of the TA generation and can be considered as an upper bound as we assumed full $CaCO_3$ dissolution above. The remaining 10 % need to be supplied by

other anaerobic, metabolic processes.

In order to estimate the contribution of denitrification as a source for TA generation in the Elbe Estuary and to shed further light on the coupling between TA and nitrogen cycles, we related imported riverine particulate organic nitrogen (PON) to metabolically released $NO_3^-$.

Without $NO_3^-$ generation by nitrification, we could explain TA generation by denitrification with around 9 % of the overall

metabolic TA gain. However, under consideration of both processes occurring in the estuary, we were unable to further attribute denitrification as the entire source for the TA generation in the Elbe Estuary. We identified a nitrogen deficit of riverine imported PON to fuel both $NO_3^-$ generation (i.e. by nitrification) and TA generation (i.e. by denitrification) in boxes 4 and 5 (Table 3) that affect the balance of the entire estuary. This in turn means that lateral $NO_3^-$ sources need to be inferred to balance the nitrogen budget. Similarly, the fairly constant $NO_3^-$ gain along the estuary (Fig. 5), but both decreasing $O_2$

consumption and DIC generation downstream (Figs. 4b, 5) imply a surplus of nitrate in the downstream part, thus a decoupling between carbon and nitrogen cycling.

To provide further evidence for lateral $NO_3^-$ sources (Kendall et al., 2007;Middelburg and Nieuwenhuize, 2001;Sigman et al., 2001), we employed nitrate stable isotope ($\delta^{15}N$, $\delta^{18}O$) signatures. Similar to Dähnke et al. (2008), minimum $\delta^{15}N$ and $\delta^{18}O$ values were found in the section with maximum observed $NO_3^-$ concentrations (box 5). The minima were lower than expected

from conservative mixing of the North Sea and river end-members and nitrification processes during the 10-30 days downstream transport of the water (see for example Spieckermann et al. (2021)) (Fig. 3a). Nitrification preferentially releases lighter nitrate, depending on the sources for nitrification as evidenced by both decreasing $\delta^{15}N$ values in $NO_3^-$ and increasing $\delta^{15}N$ values in SPM (Fig. 3a), but cannot explain the observed local minimum alone. The minimum in isotopic signature is in line with assumed delta values of the added nitrate (Eq. 2) of 7.1 ‰ for $\delta^{15}N\text{-}NO_3^-$ and -1.6 ‰ for $\delta^{18}O\text{-}NO_3^-$, respectively

(Table 4), which in turn point to an allochthones, lateral nitrate source. Such can be fueled by soil ammonium oxidation, manure and septic waste (Kendall et al., 2007), which can be identified as nitrogen sources with sufficiently light $\delta^{15}N$ and $\delta^{18}O$ characteristics. Although our study implies a lateral $NO_3^-$ source, in contrast to earlier studies (Sanders et al., 2018;Dähnke et al., 2008), we cannot elucidate the corresponding source processes and regions in the context of this study.

Within the entire estuary, the spatial distributions of $NO_3^-$ gain on one hand and the $O_2$ consumption and metabolic DIC

generation on the other hand, show incoherent patterns. We found a decoupling of carbon, nitrogen and oxygen in the middle and lower estuary, with a fairly constant $NO_3^-$ gain along the estuary (Fig. 5), but both decreasing $O_2$ consumption and DIC generation river downstream (Figs. 4b, 5). Thus, neither $O_2$ nor DIC reflect the $NO_3^-$ gain, suggesting additional nitrate sources in the downstream fresh water part of the estuary.





**Table 3. Mass balance results and POM properties.** The mass balance results are shown as metabolic generation (i.e. gains) in concentrations (mmol m$^{-3}$), in fluxes of total alkalinity (TA), dissolved inorganic carbon (DIC), and nitrate (NO$_3^-$), respectively (metabolic fluxes are also visualized in Fig. 5), and in kmol d$^{-1}$. The average carbon:nitrogen (C:N) and particulate inorganic carbon:total carbon (PIC:TC) ratios of suspended particulate matter (SPM) are given per box. The observed imported POC and particulate organic nitrogen (PON) values based on sampled filters in box 1, and the calculated transported values for PON in the other boxes (2-5) (sect. 2.3) are shown.

Imported PIC for box 1 was calculated based on imported POC, and transported PIC values were calculated for the other boxes (2-5) (sect. 2.3). DIC fueled by PIC, and TA fueled by PIC give the amount of each that can be fueled of imported and transported PIC. DIC not fueled by PIC, and TA not fueled by PIC give the amount of each that is the remaining and not fueled of imported PIC or i.e. CaCO$_3$ dissolution. The standard deviation ± sd as spatial variability is given when possible. An uncertainty estimation for analytical measurements, the air-sea flux estimation and the river discharge is given as (± absolute errors) by an error propagation (sect. 2.3). Imported POC, PIC and PON to

the estuary was estimated using measured average POC (596 ± 52 µmol L$^{-1}$), and PON (91 ± 8 µmol L$^{-1}$) concentrations of SPM (47 ± 5 mg L$^{-1}$), all determined for box 1. PIC is here a synonym for CaCO$_3$ dissolution as source.

| Species | Unit | Box 1 | Box 2 | Box 3 | Box 4 | Box 5 |
|---|---|---|---|---|---|---|
| **Metabolic TA gen.** | mmol m$^{-3}$ | - | 221 (± 3.6) | 112 (± 3.6) | 87 (± 3.6) | 1 (± 3.6) |
| **Metabolic DIC gen.** | mmol m$^{-3}$ | - | 319 (± 5.4) | 196 (± 18.8) | 138 (± 12.5) | 6 (± 4.0) |
| **Metabolic NO$_3^-$ gen.** | mmol m$^{-3}$ | - | 17 (± 0.9) | 23 (± 0.9) | 24 (± 0.9) | 20 (± 0.9) |
| **Metabolic TA flux** | mmol m$^{-2}$ d$^{-1}$ | - | 808 (± 13.3) | 76 (± 2.5) | 59 (± 2.5) | 1 (± 3.9) |
| **Metabolic DIC flux** | mmol m$^{-2}$ d$^{-1}$ | - | 1164 (± 19.6) | 132 (± 12.7) | 93 (± 8.5) | 6 (± 4.3) |
| **Metabolic NO$_3^-$ flux** | mmol m$^{-2}$ d$^{-1}$ | - | 63 (± 3.3) | 16 (± 0.6) | 17 (± 0.6) | 21 (± 1.0) |
| **Av. C:N ± sd** | - | 6.6 ± 0.1 | 7.1 ± 1.1 | 7.9 ± 0.5 | 7.9 ± 0.4 | 7.5 ± 0.4 |
| **Av. PIC:TC ± sd** | - | 0.286 ± 0.25 | 0.192 ± 5.33 | 0.294 ± 0.62 | 0.314 ± 0.25 | 0.365 ± 0.33 |
| **Metabolic NO$_3^-$** | kmol d$^{-1}$ | - | 625.3 | 857.5 | 893.5 | 723.0 |
| **Metabolic DIC** | kmol d$^{-1}$ | - | 11644.3 | 7146.7 | 5036.9 | 196.2 |
| **Metabolic TA** | kmol d$^{-1}$ | - | 8083.7 | 4090.6 | 3178.7 | 44.9 |
| **Imported POC** | kmol d$^{-1}$ | 21788.9 (± 11.4) | - | - | - | - |
| **Imported PON** | kmol d$^{-1}$ | 3322.9 (± 0.2) | 1219.7 (± 0.2) | 422.7 (± 0.2) | -756.7 (± 0.2) | -646.5 (± 0.2) |
| **Imported PIC** | kmol d$^{-1}$ | 6318.8 (± 3.3) | 5883.9 (± 3.3) | 1738.8 (± 3.3) | -1182.6 (± 3.3) | -113.8 (± 3.3) |
| **DIC fueled by PIC** | kmol d$^{-1}$ | - | 3376.8 (± 196.7) | 2072.5 (± 687.0) | 1460.7 (± 457.2) | 56.9 (± 147.7) |
| **DIC not fueled by PIC** | kmol d$^{-1}$ | - | 8267.4 (± 196.7) | 5074.2 (± 687.0) | 3576.2 (± 457.2) | 139.3 (± 147.7) |
| **TA fueled by PIC** | kmol d$^{-1}$ | - | 6753.7 (± 196.7) | 4145.1 (± 687.0) | 2921.4 (± 457.2) | 113.8 (± 147.7) |
| **TA not fueled by PIC** | kmol d$^{-1}$ | - | 1330.1 (± 237.4) | -54.5 (± 699.7) | 257.3 (± 476.2) | -68.9 (± 198.7) |

**Table 4. Added nitrate values.** The initial values and the changes of the nitrate concentration (µmol L$^{-1}$) and its associated stable isotopes (δ values in ‰) between Elbe stream km 609 and 705 are shown.

| NO$_3^-$ [µmol L$^{-1}$] | | NO$_3^-$ increase | | δ$^{15}$N-NO$_3^-$ [‰] | | | δ$^{18}$O-NO$_3^-$ [‰] | | |
|---|---|---|---|---|---|---|---|---|---|
| **609** | **705** | **[µmol L$^{-1}$]** | **[%]** | **609** | **705** | **Added δ$^{15}$N-NO$_3^-$** | **609** | **705** | **Added δ$^{18}$O-NO$_3^-$** |



| 73.9 | 164.8 | 90.8 | 122.9 | 17.1 | 11.5 | 7.1 | 8.9 | 3.1 | -1.6 |


## 3.4 Estuarine carbon release and estimated impact on coastal carbon storage

The impacts of anthropogenic and management activities on estuarine and coastal carbon cycling and $CO_2$ uptake have frequently been discussed from an aerobic perspective, as for example by Borgesa and Gypensb (2010) for the southern North Sea. Here we attempt to complement this perspective by considering anaerobic processes. We employed a biogeochemical

simulation approach (Schwichtenberg et al., 2020) to estimate the impacts of estuarine metabolic TA gains on the carbon cycle and $CO_2$ uptake capacity in the North Sea. We used this simulation approach to highlight the effect of changing DIC:TA ratios, e.g. due to anthropogenic induced changes, on the carbon storage in coastal oceans. Conceptually, the simulations reflect the relative balance between aerobic and anaerobic respiration of a given amount of organic carbon, such that any reduced alkalinity scenario replies a preference for aerobic processes, and the increased alkalinity scenarios in turn imply a preference

of anaerobic processes releasing a given amount of DIC. We applied four scenarios in which we changed the TA loads as input variable, and compared the resulting $CO_2$ uptake by the North Sea at three different distances from coast. For the first scenario, the reference run (i.e. normal condition), we used the full riverine TA and DIC load that correspond to a TA load of 100 %. For the second scenario, we reduced the TA load to 86 % reflecting the above contribution (14 %) of metabolic alkalinity to the overall alkalinity release by the Elbe Estuary. For further comparisons, we ran scenarios with a reduced TA load to 50 %

and an increased TA load to 150 %. We extrapolate these scenarios to 254 rivers that discharge into the North Sea. In particular, the major rivers and estuaries are either associated with a major port (e.g. Antwerp, Rotterdam, London or Bremerhaven), or are generally characterized as a highly turbid and heavily used waterway (e.g. Ems) that, like the Elbe Estuary, provide ideal conditions for anaerobic metabolic pathways.

Under reference conditions (100 % TA), the open North Sea (> 200 km distance from the coastline) absorbs more atmospheric

$CO_2$, with carbon transported in the deeper zones via the continental shelf pump (Thomas et al., 2004), than the coastal zones (100 km distance) (Fig. 6, absolute ASF indicated by bars). This spatial distribution is also visible in the scenarios with reduced TA loads. Assuming a remaining TA load of 50 % and 86 %, we modeled less $CO_2$ uptake due to reduced metabolic alkalinity generations.

Compared to the reduced TA scenarios, the increased TA scenario (150 %) also has higher $CO_2$ uptake within each distance,

but in spatial distribution, higher $CO_2$ uptake in coastal areas than in the open North Sea. This indicates that in the 150 % scenario the open North Sea no longer represents the area of strongest $CO_2$ uptake.

The differences in $CO_2$ uptake in the North Sea (Fig. 6, difference ASF indicated by lines) between the scenarios with reduced TA (50 & 86 %) and the reference scenario (100 %) decreased with increasing distance from the coast. This change is also reflected in the difference in $CO_2$ uptake between the scenario with increased TA load (150 %) and the reference scenario,

where the difference in $CO_2$ uptake is higher in the coastal area than in the open ocean. This phenomenon can be partly explained by mixing, as the signal is more diluted the further offshore. Furthermore, $CO_2$ equilibration continues as waters are





transported offshore (e.g. Burt et al. (2014)), i.e. away from the TA source. The larger differences in $CO_2$ uptake in coastal areas suggest that the effects of TA input from metabolic processes in rivers are similar in magnitude to the simulated (Fig. 6) and observed (Thomas et al., 2004) air-sea flux itself. Similar evidence has been presented by Burt et al. (2016), who report

that benthic sediment release in the southern North Sea shapes, if not dominates, the seasonal variability of the TA system (see also Thomas et al. (2009)).

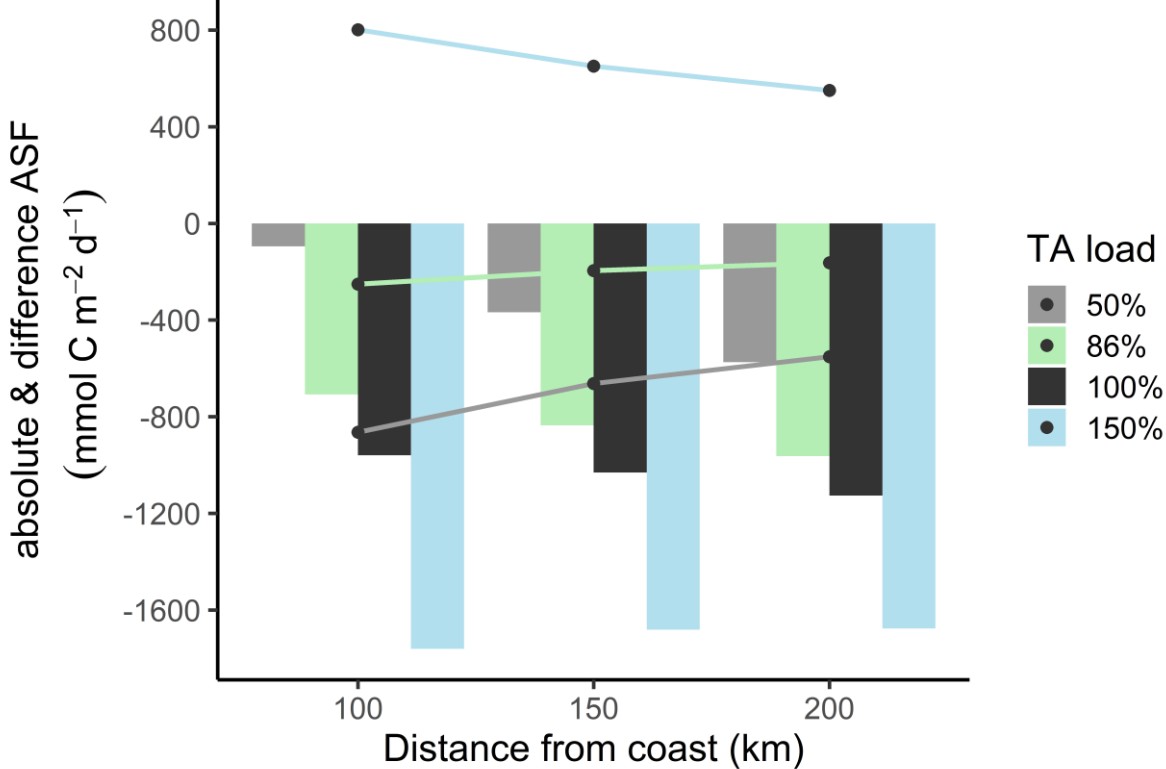

**Figure 6. Biogeochemical $CO_2$ air-sea flux (ASF) simulations at distance from coast.** The bars show the annual absolute $CO_2$ uptake in the water under reduced (50 %, grey) and (86 %, green), normal (100 %, black) and increased (150 %, blue) TA load. The lines visualize

the $CO_2$ uptake difference between the scenarios 50, 86, 150 %, and the reference scenario (100 %).

Next to $CO_2$ air-sea fluxes, the metabolic TA release from the Hamburg port area could also affect calcifying organisms such as foraminifera that occur in the lower Elbe Estuary. In a recent study, Francescangeli et al. (2021) observed the change of calcite saturation state ($\Omega_{Ca}$) from under- to supersaturation around Elbe km 685 (approx.). Without the respective TA

generation, the supersaturated range and thus the habitat of the foraminifera, would clearly be further downstream, if not restricted to the North Sea. For example, at their most saline station (Francescangeli et al. (2021), see their Fig. 2f), corresponding to Elbe stream km 730 in our study, the observed $\Omega_{Ca}$ is ~ 3.3, whereas it would be strongly undersaturated ($\Omega_{Ca}$ ~ 0.2) without benthic TA release.





## 4 Conclusions

We observed clear regional differences in the biogeochemistry of the Elbe Estuary. While conservative mixing prevails in the salinity gradient of the lower estuary, metabolic processes dominate the upper, fresh water part with the port of Hamburg. We observed a strong increase of several hundred $\mu mol \ kg^{-1}$ of TA and DIC in the Hamburg port (Fig. 3) with vanishing values downstream, and calculated an annual metabolic TA load of about 14 % of the total TA runoff from the Elbe River to the coastal ocean.

Mass balances suggest that up to 90 % of the generated TA of the entire estuary could be fueled by imported PIC, i.e. through $CaCO_3$ dissolution. Anaerobic metabolic processes such as denitrification, iron, manganese, or sulfate reduction could fuel the remaining 10 % of TA generation. Accordingly, estuarine ecosystems appear to be highly susceptible in particular to nutrient management actions that affect the balance of the various metabolic processes to generate TA by controlling $NO_3^-$ availability, as organic carbon and oxygen supply appear in excess or steady, respectively. One could speculate that the reduced organic

carbon supply would reduce the importance of anaerobic processes in favor of a relative increase in aerobic respiration. On the other hand, a reduction in $NO_3^-$ for a given organic carbon content would trigger other anaerobic processes (e.g iron or sulfate reduction) that have a higher TA gain per unit of carbon respired than does denitrification (Chen and Wang, 1999). The interplay of the various metabolic processes governs both the release of reduced products and the carbonate system properties in the estuary. For example, the former controls the balance of released gases such as $N_2$, $N_2O$ and $H_2S$, which in

turn can have negative effects on the entire estuarine biodiversity, leading to losses in reproduction and habitats of e.g. local fish species (Breitburg et al., 2009;Wu, 2009;Breitburg, 2002;Janas and Szaniawska, 1996). The carbonate system properties regulate the habitable range for example for benthic foraminifera(Francescangeli et al., 2021). We show that the metabolic TA release increases capacity and $CO_2$ absorption in the German Bight. We postulate that this result may be transferable to other global rivers or estuarine systems, particularly those with large port facilities, and that it could have tangible implications for

the ocean's $CO_2$ uptake capacity on a global scale.

### Data availability

The datasets generated during and/or analyzed during the current study are either presented in the study, and are in preparation to be released in accordance with the rules of the funding agency.

### Author contributions

MN, KD, and HT designed the study. MN did the sampling, sample measurements and analyses, data interpretation and evaluation, and prepared the manuscript. JP did the biogeochemical simulation by using the 3D-ECOHAM model. GS performed stable isotope measurements, data analysis and interpretation, and added to the method description. KD, TS, JP, JEEVB, and HT provided scientific and editorial recommendations. MN wrote the manuscript with input from all co-authors.





**Competing interests**

The authors declare that they have no conflict of interest.

**Acknowledgement**

This research has been funded by the German Academic Exchange Service (DAAD, project: MOPGA-GRI, grant no. 57429828), which received funds from the German Federal Ministry of Education and Research (BMBF).

We thank our crew of the RV *Ludwig Prandtl* for their helping hands during the sampling campaign. Thanks to Leon Schmidt

for measuring the nutrients, our intern Jeannette Hansen for supporting us with the box model calculation, Linda Baldewein for supporting us with the Elbe River kilometer calculation, the working group of biogeochemistry at the Institute of Geology, University Hamburg for measuring the filters, and Yoana Voynova for providing the FerryBox data. The oxygen calibration samples were collected by Götz Flöser, analyzed by Tanja Pieplow, and processed by Yoana Voynova. The FerryBox preparation (optode specific) was done by Martina Gehrung.

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
