# Peer review of "Metabolic alkalinity release from large port facilities (Hamburg, Germany) and impact on coastal carbon storage"

_Biogeosciences, 2022_

## Author Response (AR1)

**Editor Comments to the author**:
As you can see, both reviewers are positive about your manuscript, but have a number of concerns, which you have identified and addressed.

Picking up on a main point for both reviewers, the figure of 90% TA generation from CaCO3 dissolution is quite uncertain and is based on the PIC:POC ratio entering and assuming all PIC dissolves (and also all POC is mineralised). I would like to see this considered and discussed a little more. This assumption suggests there should be no CaCO3 accumulation in the harbour sediments. I would have thought there is some data on this that could help verify this?

Also how do you rule out sulfate reduction and burial of sulfides as a significant source of TA?

Please revise your manuscript as you have outlined as well as taking my comment above into consideration.

*AC: Dear Mr. Cook,*
*Thank you very much for your positive response.*
*We thank you and the two anonymous reviewers for taking the time to review our manuscript. All the insightful and constructive comments were great and very helpful in improving the manuscript.*
*In accordance with your suggestions above, we have discussed CaCO3 formation in more detail and added a statement on sulfate reduction in section 3.3.*
*Below are the point-by-point responses to the reviewers.*

*Yours sincerely,*
*On behalf of all co-authors,*
*Mona Norbisrath*

**BG-2022-143 RC1**

**General Comments**

This well written and thorough manuscript describes the biogeochemical processes occurring in the Elbe river estuary and links these to anthropogenic influences. The modelled biogeochemical cycles are well considered and describe a large source of TA within the estuary. This TA source is described as being caused by the dissolution of CaCO3 sediments which is assumed to be driven by the increased organic matter and N loading of the estuary. Biogeochemical processes are explored in detail, although much of this is hypothetical discussion.

One weakness of the study is that all biogeochemical processes are estimated from dissolved concentrations rather than from other field observations, for example benthic sediment incubations or water profiles. The estimation of biogeochemical processes within estuaries is complex due to the dynamic interactions occurring over tidal cycles (salt wedge and tidal movements influencing benthic sediment interactions), diel cycles (changes is benthic O2 caused by respiration/photosynthesis), seasonal cycles influencing metabolism rates, and all influenced by different riverine flow rates. While I agree that the processes identified here are likely occurring within the estuary, the accuracy of the predictions (e.g. that 90% of TA is due to CaCO3 dissolution) may have a high level of inaccuracy and may have high temporal variability. It is useful to use modelled results, however it is important to highlight the uncertainties and assumptions associated with them in all sections of the manuscript.

*AC: Dear reviewer, thank you very much for your positive feedback and the helpful comments and suggestions to improve the manuscript. We have addressed your suggestions in order to improve the manuscript. You will find our answers below. The referred lines belong to the manuscript version that includes the track changes.*

*We used field-observations of dissolved inventories that include the net metabolic generated amount of the parameters, rather than rate measurements. We rely on the strength of integrative capacities of our tracers (TA & DIC), which facilitate reliable estimations of metabolic processes. Such estimations should be seen as a powerful approach complementary to direct rate or process assessments.*

*We are aware that this natural system is variable and naturally influenced. The assessment of variability is out of scope of this work, but the further investigation of natural variability should be addressed in future research. We added a statement to uncertainties due to natural variability in the text. The measurement uncertainties are included in the error propagation.*

**Other Comments**

L17 – wording - 'resulting in maximum'

*AC: We changed the wording in this sentences "resulting in a maximum TA generation of 90% due to...". (L17)*

L60 – I suggest separating the two research questions for clarity. I also note that the 2$^{nd}$ research question doesn't receive much attention in the abstract or conclusion.

*AC: We included sentences with highlighting the result of the second research question in the abstract (L21-22) and conclusion (L505-506), and highlighted the two research questions in the introduction (L63ff).*

L65 – Provide more detail on 'surface water samples'.

*AC: We added more details for the surface water samples and how they were collected (L85ff).*

L139 – What is the source of wind speed measurement DWD? If wind speed was not measured in situ then where was the data source and at what resolution was the data collected as wind speed can greatly influence flux estimates and this uncertainty should be clarified.

*AC: The wind speed was measured in situ by the federal authority Deutscher Wetterdienst (DWD). We clarified this in the text (L167-168).*

L157 – Provide details of box volume measurements and fill time estimates. Errors associated with both volume and fill time should be incorporated into the mass balance calculations. E.g. errors in the generic river width, depth etc. in Table 1.

*AC: For the box separation, we defined boxes that of course does not consider the real river volume. However, the uncertainty of the box volume is indirect already included in the error estimation while estimating the error of fill time, as the product of box volume and discharge.*

L176 – Outer boundary conditions measured two months later. Is this an issue? Please justify.

*AC: We do not consider the use of the outer boundary conditions as an issue, because they represent average summer values for this region in the North Sea. The data originated from the nearest observation to point and time, and the difference in observational time appears to be much smaller than the flushing time of the southern North Sea.*

L190 – The sentence is vague and poorly structured. Clarify the term 'imported' and clarify the link between PIC and CaCO3 dissolution.

*AC: Thank you for this comment. We restructured the sentences and clarified the term "imported" and the link between PIC and $CaCO_3$ in the text (L218ff).*

L308 – This section contains the most clear findings of the paper but seems hidden within the manuscript.

*AC: Thank you for highlighting this. We restructured the section and removed the text interruptions by Table 2 and Figure 5, while putting the text together, to highlight its importance more. (L295-341)*

Table 3. Errors of species in the first half of the table seem optimistically low. Are all cumulative errors considered? Errors of samples, box volumes, flow rates, fluxes…etc.

*AC: Yes, we applied a standard error propagation, thus accumulated all errors. The absolute errors are listed in Table 3.*

L437 – replace 'vanishing' with a more appropriate term.

*AC: We changed vanishing into diminishing. (L493)*

L442-L447 – This speculation reads as discussion not conclusions

*AC: We agree and replaced this part into the discussion section 3.3.*

END OF COMMENTS

**BG-2022-143 RC2**

**General Comments**

First, I would like to congratulate the authors on a very interesting manuscript. It provides insight into the biogeochemistry of the described estuary and attempts to quantify contributions of alkalinity from various sources.

I have a couple general comments about this manuscript. Firstly, I would like to suggest that the authors have the manuscript thoroughly proof-read before publication. I found several grammatical issues and a general lack of flow due to sentence/paragraph structure throughout the manuscript. In the section below I have identified some of these. This should be addressed for the reader's benefit.

Next, I have a general comment about the methods section. I think the authors should consider restructuring this section. First, a detailed description of the study site should be presented. I found that the authors did not go into enough detail when describing the estuary and it would really help "set the scene" for the reader if they bulked this up a bit. Sampling protocols should be removed from the study site section and have its own section that follows that describes the water sampling techniques as well as how each individual parameter was subsampled and preserved. The model/make of the CTD + O2 probe on the FerryBox should also be provided. I noticed that the authors have grouped sampling/preservation techniques in with each analytical procedure. These should be removed and added to the new sampling section above to make the process of the study easier to follow sequentially. Try the following structure for the methods: Study Site -> Sampling -> Analytical Procedures…etc.

My final comment very similar to that of Reviewer 1. The processes described in this manuscript are based on estimations instead of field observation. On top of that, the dynamic nature of estuaries can lead to high variability both spatially and temporally which adds large uncertainties to estimations from models. Although the conclusions presented by the authors are compelling, they should add a note in the discussion or conclusions stating that these estimations may be serendipitous due to large levels of uncertainty.

*AC: Dear reviewer, we gratefully appreciate your helpful comments to improve the manuscript. We have considered your comments and reworked the manuscript to improve it. You will find our answers below. The referred lines belong to the manuscript version that includes the track changes.*

*In accordance to your suggestion, we restructured the Methods section and added a more detailed study site description as section (2.1). In there, we also included some sentences, which were previously in the Introduction and removed them there to prevent doubling. We also added the information of the O2 optode. A more detailed description of the FerryBox is given in the reference (Petersen et al., 2011) which is given in the text. We also added a statement to uncertainties due to natural variabilities in the text (L377, 498). Also, a native speaker thoroughly proof-read the manuscript and we reworked the grammar and structure in the entire manuscript.*

*We used field-observations of dissolved inventories that include the net metabolic generated amount of the parameters, rather than rate measurements. We rely on the strength of integrative capacities of our tracers (TA & DIC), which facilitate reliable estimations of*

*metabolic processes. Such estimations should be seen as a powerful approach complementary to direct rate or process assessments.*

*We are aware that this natural system is variable and naturally influenced. The assessment of variability is out of scope of this work, but the further investigation of natural variability should be addressed in future research. We added a statement to uncertainties due to natural variability in the text.*

**Other Comments**

L31: Add Gilbert et al. 2005 (https://doi.org/10.4319/lo.2005.50.5.1654) as a reference when describing oxygen deficits. It gives a 72-year record of oxygen depletion in the St. Lawrence Estuary.

*AC: Done (L32).*

L34: Grammatical error – "It encompassing an area that begins" should read "It encompasses an area that begins".

*AC: Done. We also restructured the sentence (L69).*

L39-41: The following piece of writing does not flow well: "Eutrophication can cause large phytoplankton blooms both in rivers and in the coastal zone. Whose decay increases oxygen consumption….". These few lines should be re-written to read something like the following "Eutrophication can generate phytoplankton blooms in both rivers and the coastal zone. This increased flux in organic matter can lead to higher rates of oxygen consumption which can drive hypoxia in stratified bodies of water such as estuaries and the coastal ocean".

*AC: We thank you for this comment. We have rearranged the sentence in accordance with your comment (L41ff).*

L41-42: When describing hypoxia in the coastal ocean you should add the following sources: Gilbert et al. 2005 (https://doi.org/10.4319/lo.2005.50.5.1654), Rabailas et al. 2001 (https://doi.org/10.2134/jeq2001.302320x), and Rabailas et al. 2002 (https://doi.org/10.1146/annurev.ecolsys.33.010802.150513).

*AC: We have included the suggested references in the text (L46ff).*

L42-44: Again, there are some issues with flow here. Instead of describing the dredging then having a very short sentence with a statement that organic matter turnover exists, you should combine the sentences. Try something like: A hotspot of organic matter turnover exists upstream of the Port of Hamburg, where recently the sea floor has been dredged (increasing the depth from ~5 m to ~20 m) to increase accessibility for ships."

*AC: According to your comment, we have rearranged the sentences (L46ff).*

L57-60: I agree with Reviewer 1 on this section. The goals of this study should be separated for clarity. Also, the authors may want to consider removing the use of first person identifiers such as "We" or "I" in the manuscript and opt for a third person approach. For example changing "We want to answer the questions of…." to "This study aims to answer the questions of…".

*AC: We clarified the aims of the study by including a) and b) (L63ff). We changed the "This study aims to.." sentence in the Introduction like suggested, but we would like to stick to the first person approach in the major part of the manuscript.*

L65: Grammatical error – "This study based on samples…." should read "This study is based on samples…".

*AC: Done (L83).*

L155-157: Please provide individual box volumes and fill times.

*AC: Individual box volumes and fill times are provided in Table 1 for each box. To make this more clear, we referred to Table 1 in the text.*

---

## Author Response (AR2)

**Editor Comments to the author**:
Thank you for revising your manuscript. I am still unsatisfied with your justification for partitioning the alkalinity fluxes as you have. I had a look at Amann et al (2015) and it seems undersaturation of calcium carbonate is not a universal phenomenon in the Elbe. I also note Amman et al argue sulfate reduction is not likely to contribute to alkalinity production due to course sediments and an oxic water column. Without further data, however, I remain unconvinced by this argument. In an environment with high colloidal inputs and low salinities it seems highly likely that sulfate reduction and the formation of FeS will occur, which leads to alkalinity generation. Are there no areas of black sediment (indicating acid volatile sulfides) accumulation within the Port? In the absence of further information, I suggest you can only discuss likely sources qualitatively, rather than your current apportionment.

*AC:*
*Dear Mr. Cook,*
*Thank you for your thoughts on the revised manuscript.*
*We understand your concerns and have decided to significantly tone down this discussion, as in fact our data set does not allow us to draw any substantiated conclusions here. For this reason, we prefer to make neither favorable nor unfavorable statements.*
*We changed the text to:*
*"This result can be supported by the studies of Kempe (1982) and Francescangeli et al. (2021), who report undersaturated calcite saturation states (Ω) in the upper estuary and at their most fresh water station in the middle estuary.*
*Our estimate should be considered an upper bound, since other anaerobic metabolic processes must provide the remaining at least 10 % of the generated TA we observed. However, our data set does not allow us to directly identify or even quantify any of these processes, nor does it allow us to exclude them. "*

*In this context please allow us to point to a factual error in the paper by Amann et al. (2015), namely their statement on page 205 on line 1 that supersaturation occurs if the omega is larger than zero. Obviously, the saturation state of omega indicates supersaturation with values larger than one.*

*Yours sincerely,*
*On behalf of all co-authors,*
*Mona Norbisrath*

---

## Author Response (AR3)

**Editor Comments to the author:**
Thank you for your response to my comments. I don't think you really understood the breadth of my request so I will try to be a bit more explicit.

Your approach for alkalinity source attribution in section 2.5.2 is not robust (although I am ok with the denitrification derived estimates) as discussed in previous communication. Just to recap for clarity. Although it is likely that your alkalinity is derived from CaCO3 dissolution, CaCO3 dissolving is likely to have been historically deposited in the sediment and does not necessarily reflect what is currently entering as you account for in your box model. Furthermore, it seems possible that there are regions of FeS generation and TA production not accounted for in your model. Given these uncertainties, I don't think you can add the calculations you have to the box model.

I suggest you remove (or heavily edit to remove the CaCO3 estimates) section 2.5.2 and use the discussion to argue where you believe the alkalinity is derived from. You will also need to change any text where you apportion alkalinity eg line 464.

*AC: Dear Mr. Cook,*

*Thank you very much for your in-depth thought with regard to our manuscript. We feel we have addressed the referee comments, in particular addressing other metabolic sources of alkalinity and temporal variability in our revised version. As such, admittedly, we have problems following, in part even identifying your concerns. Please allow us to comment below:*

*1: Necessary and fundamental condition for any box model approach is steady-state, thus any argument of temporally varying processes/fluxes cannot be entertained by such an approach. This fundamental condition is clearly stated in equation 6. Furthermore, in our discussion we have picked up this point, amongst other sections 3.3. and conclusions.*
*However this fundamental condition requires that per time interval (or over the integration time of the model) inputs and outputs balance each other. As such we cannot consider dissolution of "historic" CaCO3, as this would violate the steady-state condition.*
*On the other hand for our approach it is irrelevant whether CaCO3 dissolves in the water phase or in/on the sediments, as long as we overall assume that inputs balance outputs per time step.*

*2: Furthermore, and as importantly, our approach pursues a two-step strategy.*
*First we establish dissolved mass balances and quantify a gain/loss of certain properties, such as alkalinity DIC and nitrate.*
*This approach relies on our observations with some clearly defined baseline conditions such as box volume, discharge etc., all of which have been clearly identified in our paper. The only, but crucially important outcome of this step are above mentioned terms of respective losses or gains – this without any attribution of responsible processes. We perpetuate this strategy in section 3.4 (methods 2.6), where we only consider above gain and loss terms, without any further source attribution.*

*In our second step we propose a source attribution to these processes. This is based on further observed variables, such as (directly observed) properties of POM, in detail the POC:PIC ratio, the POC:PON ratio etc. Here we show that it is possible to explain up to approx. 90% of the TA gain by dissolution of the imported CaCO3, this with clear reference to above steady-state assumption. We explicitly state that this is to be considered as an upper bound, if we assume complete dissolution of imported CaCO3. We also rearranged the wording in the sentence starting in line 15 to make the upper bound approach more clear.*
*We do not exclude and cannot exclude any other sources of alkalinity, and attribute to these source, such as denitrification, combined the remaining gain of TA, as a lower bound. Obviously these upper and lower bounds, respectively, depend on the dissolving proportion of the imported CaCO3, again invoking steady-state conditions.*

*In summary, we feel that we have addressed both your and the overall positive referee comments in latest revised version "on the spot" and comprehensively. Furthermore, we also feel that we have discussed limitations, such as aspects of temporal variability or the role of other TA sources openly during the later section of the manuscript. We therefore hope that our manuscript is suitable for publication in BG now.*

*Yours sincerely,*
*On-Behalf of all co-authors,*
*Mona Norbisrath & Helmuth Thomas*